# Eye and Hair Color Prediction of Ancient and Second World War Skeletal Remains Using a Forensic PCR-MPS Approach

**DOI:** 10.3390/genes13081432

**Published:** 2022-08-12

**Authors:** Irena Zupanič Pajnič, Tomaž Zupanc, Tamara Leskovar, Matija Črešnar, Paolo Fattorini

**Affiliations:** 1Institute of Forensic Medicine, Faculty of Medicine, University of Ljubljana, Korytkova 2, 1000 Ljubljana, Slovenia; 2Department of Archaeology, Faculty of Arts, University of Ljubljana, Zavetiška 5, 1000 Ljubljana, Slovenia; 3Department of Medicine, Surgery and Health, University of Trieste, Strada per Fiume 447, 34149 Trieste, Italy

**Keywords:** PCR-MPS, phenotyping, DNA degradation, PCR artifact

## Abstract

To test the usefulness of the forensic PCR-MPS approach to eye and hair color prediction for aged skeletons, a customized version of the PCR-MPS HIrisPlex panel was used on two sets of samples. The first set contained 11 skeletons dated from the 3rd to the 18th centuries AD, and for each of them at least four bone types were analyzed (for a total of 47 samples). In the second set, 24 skeletons from the Second World War were analyzed, and only petrous bones from the skulls were tested. Good-quality libraries were achieved in 83.3% of the cases for the ancient skeletons and in all Second World War petrous bones, with 94.7% and 100% of the markers, respectively, suitable for SNP typing. Consensus typing was achieved for about 91.7% of the markers in 10 out of 11 ancient skeletons, and the HIrisPlex-S webtool was then used to generate phenotypic predictions. Full predictions were achieved for 3 (27.3%) ancient skeletons and 12 (50%) Second World War petrous bones. In the remaining cases, different levels of AUC (area under the receiver operating curve) loss were computed because of no available data (NA) for 8.3% of markers in ancient skeletons and 4.2% of markers in Second World War petrous bones. Although the PCR-based approach has been replaced with new techniques in ancient DNA studies, the results show that customized forensic technologies can be successfully applied to aged bone remains, highlighting the role of the template in the success of PCR-MPS analysis. However, because several typical errors of ancient DNA sequencing were scored, replicate tests and accurate evaluation by an expert remain indispensable tools.

## 1. Introduction

Capillary electrophoresis (CE) analysis of PCR (Polymerase Chain Reaction) products is the gold standard in forensics [1], both for personal identification and for predicting externally visible characteristics (EVCs) such as eye color, hair color, and skin pigmentation [2,3,4,5,6]. In addition, the introduction of PCR-MPS (massive parallel sequencing) techniques [7,8] makes possible the development of EVC tools that were inconceivable until a few years ago at accessible costs [9]. Similarly, to help identify human remains or to provide investigative leads in cold cases such as homicide or rape, several PCR-MPS kits allowing bio-geographical ancestry (BGA) determination were developed in the last few years both for Illumina [10,11] and Ion Torrent technologies [12]. The ability of reporting on the EVC, BGA and age based on DNA obtained from biological samples makes the exploration of such markers and the development of new methods meaningful for criminal investigations, and the VISible Attributes through GEnomics (VISAGE) Consortium fully optimized and validated the VISAGE appearance and ancestry tools for forensic casework implementation [13,14].

The availability of EVC data is not only of interest for investigative purposes in forensics [1], but it can also be of great help in solving historical and archaeological issues [15,16,17,18,19,20,21,22]. Therefore, cooperation among geneticists, archaeologists, anthropologists, and historians has been encouraged [23]. There are several aspects that forensic DNA analysis and studies of ancient DNA (aDNA) have in common, such as the limited amount of highly degraded DNA, the precautions that have to be followed to avoid contamination, and the adoption of authenticity criteria to confirm endogenous bone DNA [24]. In aDNA studies, the PCR-based approach has recently been replaced to a large extent with new techniques leading to the typing of even entire genomes [25,26]; nevertheless, in comparison to forensic techniques, they are much more expensive, and they require well-equipped laboratories and well-trained staff.

With this in mind, the goal of this study was to test the usefulness of the forensic PCR-MPS approach in eye and hair color prediction for aged skeletons. In particular, a panel of EVC markers was typed in ancient remains using a set of different skeletal element types from each skeleton and tested to determine whether a reliable prediction can be achieved through generation of a consensus genotype. For more recent samples from the Second World War, it was tested whether a single test-based approach, with no repetition of analysis [27], could lead to reliable EVC typing. Only one skeletal element type that is the highest DNA-yielding bone type in the human body—the petrous bone—was selected for this purpose. The petrous portion of the temporal bone, in fact, has been shown to be the most suitable skeletal part [28] for sampling ancient skeletons [29,30], and it has also been used successfully in forensic genetic investigations [31,32,33,34].

Results are presented for the use of a customized version of the HIrisPlex panel (permitting eye and hair color prediction) on a set of 47 bone samples belonging to 11 ancient skeletons dated from the 3rd to the 18th centuries AD and on a set of 24 skulls found in Second World War mass graves.

## 2. Materials and Methods

### 2.1. Bone Samples

Eleven nearly complete ancient skeletons excavated in Slovenia from 1984 to 2017 were considered in this study. All the skeletons were discovered in non-disturbed individual graves, and after excavation the bone remains were washed with water, air-dried, and analyzed by a physical anthropologist. Based on the archaeological context and anthropological analyses, all the bones in each grave were attributed to one individual. After anthropological analysis, the remains were placed in a cardboard box and kept in a museum storeroom under dark, dry, cold, and stable conditions until this study. These skeletons were dated in line with historical and anthropological records from the 3rd to 18th centuries AD (see Appendix A for details). From each of the 11 skeletons, 11 to 21 bones and up to 5 teeth were initially used for DNA extraction, whereas for HIrisPlex sequencing, 47 DNA samples that produced the highest DNA yields, as assessed by qPCR, were selected (four to five DNA samples per skeleton; see Appendix A).

In addition, a set of petrous bones from 24 skulls belonging to Second World War victims (approximatively 75 years old) and excavated from various Slovenian mass graves [35] was also included in the study and used for DNA extraction.

### 2.2. DNA Extraction

All procedures were performed in rooms dedicated solely to processing aged bone remains, and well-established procedures to avoid contamination [36,37,38,39,40] were followed as reported in Appendix B.

The bones and teeth were ground into powder for 1 min at 30 Hz using a homogenizer (Bead Beater MillMix 20, Tehtnica—Domel, Železniki, Slovenia). To prevent bone warming during drilling and cutting, a lower speed setting was used for abrasion and cutting, and the bones were frequently cooled in liquid nitrogen. In addition, grinding vials and bone and tooth samples were cooled in liquid nitrogen before grinding. The powdered samples were weighed, and 0.5 g of bone and tooth powder from each of them was used for DNA extraction. Decalcification and purification of DNA were performed following the procedure described previously [41], with a final elution volume of 50 µL. Extraction-negative controls (ENC) were carried out in parallel [36,37,38,39,40].

### 2.3. DNA Quantification

DNA quantification was performed by qPCR. The PowerQuant^TM^ kit (Promega, Madison, WI, USA) was used at the suggested conditions for each sample in duplicate in line with the technical manual [42]. Raw data were obtained using the ABI 7500 Real-Time PCR System (Applied Biosystems—AB, Foster City, CA, USA), and the results were converted using PowerQuant Analysis^TM^ software (www.promega.com/resources/tools/powerquantanalysis-tool, accessed on 28 January 2018). The imported data were then used to evaluate standard curves (acceptable *r*^2^ and slope values), to determine quantities of Auto (Autosomal), Deg (Degradation), and Y targets in the samples, and to calculate the [Auto]/[Deg] ratio together with the IPC (Internal Positive Control) shift value (see Appendix A).

### 2.4. HIrisPlex Sequencing on Ion S5

HIrisPlex sequencing on Ion S5^TM^ (ThermoFisher Scientific, Waltham, MA, USA) was used for individual tests of each of the 47 ancient samples and petrous bones from all 24 Second World War skulls (see Appendix A). The one exception was ancient sample 7D, which was tested in duplicate.

For amplification, the PCR primer couples described in the original studies [9,27] were used. Appendix A shows the panel of the 24 SNPs that can be typed by sequencing the 17 amplicons produced by a single round of multiplex PCR. Out of these 24 SNPs, 23 are bi-allelic SNPs, whereas rs796296176 (already named rs312262906) is an InDel marker (https://www.ncbi.nlm.nih.gov/snp; accessed on 10 April 2022). As shown in Appendix A, PCR was run at the recommended number of 25 cycles for 0.3 to 1.0 ng of the sample, 27 cycles for 0.1 to 0.3 ng of the sample, and 28 cycles for less than 0.1 ng of the sample [43]. DNA concentration was determined in each sample as assessed by the Auto target of the PowerQuant^TM^ kit in duplicate quantification tests. Negative template controls [NTCs] and ENCs were tested along with the experiment.

Fully automated library preparation was performed using the Precision ID DL8 Kit^TM^ for Chef, and 32 libraries were combined into one tube for both Ion 520^TM^ and Ion 530^TM^ chips in line with the manufacturer’s user guide [43]. The concentration of the combined library pool was determined in duplicate together with standards and negative control by qPCR with the Ion Library TaqMan Quantification Kit^TM^ (TFS) [44]. Three library pools (30 pM combining 32 samples) were used for fully automated DNA template preparation on the Ion Chef^TM^ System. The templates were prepared using the Ion S5 Precision ID Chef^TM^ Reagents and loaded using the single chip loading workflow. Sequencing made use of Ion S5^TM^ Precision ID Sequencing Reagents and Ion S5^TM^ Precision ID Sequencing Solutions [44].

### 2.5. Sequencing Data Analysis and Genotyping

The alignment of reads against the *Homo sapiens* reference genome (hg19) was performed using Ion Torrent^TM^ Suit Software 5.6 (TFS). Coverage analysis was carried out with the Coverage Analysis v 5.6.0.1 plugin, which provides statistics and graphs describing the level of sequence coverage produced for targeted regions. Information about mapped reads, on-target percentage, mean depth, and uniformity of coverage were downloaded for each sample library (Barcode Summary Report file). The resulting Excel files were then used for data analysis, and the relative depth of coverage (rDoC) was calculated as the ratio between the base coverage for a specific SNP and the overall base coverage of the sample. To compare the rDoC between the two sets of samples (e.g., ancient skeletons and Second World War bones), an *r*^2^ test was carried out.

For genotyping, the Converge^TM^ software version 2.0 (TFS) was used, applying the manufacturer’s default settings. In particular, a minimum coverage of 20× is required for genotyping, with each strand with more than 10× of coverage. A MAF (major allele frequency) flag is assigned by the analysis software to heterozygous genotypes when the reads of the two alleles are unbalanced (10.1–35.0% or 65.0–89.1%), whereas homozygous genotypes are alerted when reads corresponding to a second allele account for 1.0% to 10.0% of the entire coverage of the marker. In addition, the locus-to-locus thresholds values suggested by Breslin et al. [9] and Kukla-Bartoszek et al. [27] were applied. Therefore, the occurrence of NN typing as well of other artefacts (allelic imbalance, allelic dropouts, etc.) is given out of the number of markers above the threshold values shown in Appendix A. Finally, because eight loci showed a tri-allelic pattern, the BAM files of those samples were independently analyzed by the use of a different bioinformatic workflow. To this end, Samtools View software (version 1.7) [45] was used to identify the fragments spanning the polymorphic regions. The coverage of each SNP position was then computed by the use of in-house script software created by Python (version 3.4.1) [46].

Eight libraries that showed very low levels of mapped reads and yielded no genotype were submitted to further analysis to verify the occurrence of out-of-target sequencing. To check for the presence of overrepresented reads, the BAM files of these samples were analyzed on FastQC version 0.11.8 [47]. They were then downloaded and analyzed with Clustal W version 2.1 (Larkin, M.A. 2007). The output of the sequence alignment was visualized with Jalview 2.11.0, and, to determine regions of similarity with bacterial and fungal sequences, the most frequently represented ones were compared on BLAST [48].

### 2.6. Generation of Consensus Genotype and Eye/Hair Color Prediction

When genotyping skeletal remains, the generation of a multi-sample consensus genotype is advisable if the number of available skeletal elements permits it [25,36,40,49]. To build the consensus genotype of each of the 11 skeletons, the method recently described by Turchi et al. [50] was adopted. In particular, homozygosity was assigned only if all the tests (from the different bone elements of the same skeleton) yielded the same homozygous pattern. Heterozygosity (Ht) was assigned in the following cases: (1) two (or more) identical heterozygous genotypes, (2) one heterozygous and two (or more) homozygous genotypes for different alleles (in agreement with allelic drop-out events), or (3) two (or more) homozygous genotypes for different alleles (in agreement with a heterozygous genotype). In the case of one heterozygous plus two (or more) identical homozygous genotypes (e.g., A/G plus A/A+ A/A+ AA), no available (NA) data were assigned. NA was also assigned when the data were available from only two samples that had different genotypes at the same marker (e.g., A/G and A/A).

For the phenotypic prediction, the consensus profiles were first converted into the corresponding input codes by using the R-script program of the webtool at https://walshlab.sitehost.iu.edu (accessed on 5 March 2021). Thereafter, the resulting codes were uploaded into the webtool at https://hirisplex.erasmusmc.nl (accessed on 20 April 2021). The same procedure was followed for the 24 Second World War petrous bone samples. For these 24 samples, as well as for sample #192 (i.e., the unique sample that provided data from sk_16), the markers flagged for allelic imbalance phenomena were treated as NA data.

### 2.7. STR-CE Typing

STR typing was performed only on eight samples that yielded no result by HIrisPlex PCR-MPS typing (see Appendix A). For all of the samples except for sample #82 from sk_3, no less than 0.5 nanogram of template was amplified using the PowerPlex ESI 17 System (Promega) at standard conditions of 30 cycles. For sample #82, 0.2 nanogram of DNA was amplified. CE analysis was performed using an automatic ABI PRISM 3130^TM^ Genetic Analyzer (Applied Biosystems, Waltham, MA, USA). Negative controls were tested simultaneously.

### 2.8. Elimination Database

An elimination database was constructed to check for authenticity of genetic profiles obtained from old skeletal remains; this allows traceability in the case of contamination. For the elimination database, buccal swabs were obtained from everyone recently engaged in handling and analyzing ancient skeletons and from all the individuals that participated in the exhumation and subsequent anthropological and genetic analyses of the Second World War skeletal remains. For extraction of DNA from buccal smears on sterile cotton swabs, the EZ1 DNA^TM^ Investigator Kit (Qiagen, Hilden, Germany) was used. STR typing was performed using the NGM^TM^ amplification kit (TFS).

### 2.9. Calculations and Graphs

Microsoft^(R)^ Excel^(R)^ 2007 and Stata 16^TM^ (Stata Corporation, 2017, College Station, TX, USA) were used for calculations and graphs. For all statistical analyses (two-tailed *t*-test and ANOVA, when appropriate), significance was assumed with *p*-values *<* 0.05.

## 3. Results

### 3.1. DNA Quantification

Altogether, 71 DNA samples were considered (47 samples from the 11 ancient skeletons and 24 samples from the Second World War skulls). In total, 4 to 5 DNA samples, obtained from different skeletal element types, were used for each ancient skeleton (see Appendix A, which shows the average quantification data of duplicate qPCR tests).

Figure 1 shows the quantity of DNA extracted from each ancient skeleton as assessed by the 84 bp-long Auto probe. DNA quantity varied broadly from skeleton to skeleton (*p*-value = 0.0012), with no relation (*r*^2^ = 0.303; *p*-value = 0.079) to the dating of ancient remains. In addition, the total average DNA yield (4.3 ng of DNA/gram of bone powder) of 47 ancient skeletal elements differed from the total average DNA yield (129.5 ng of DNA/gram of bone powder) of the 24 petrous bones from Second World War skulls (*p*-value = 1.8 × 10^−5^). Lastly, the amounts of DNA detected by the two Y-specific probes (81 bp and 136-bp-long, respectively) were always lower, and poorly related (*r*^2^ = 0.247) with those of the Auto probe.

Even the DNA degradation level varied widely among the 11 ancient skeletons, with sk_4 (18th century) and sk_16 (4th century) always showing un-calculable Auto/Deg ratios because of the lack of amplification of the 249 bp-long Deg amplicon (see Appendix A). In total, 20/47 ancient samples (43.5%) exhibited this undesirable feature. Conversely, the Auto/Deg ratio was always computed in the 24 petrous bones from Second World War skulls (average = 99.8; median value = 88; min = 8.2; max = 334), indicating lower degradation levels in Second World War skeletal remains.

### 3.2. DNA Sequencing

The samples were run in three different chips, whose main sequencing parameters are shown in Appendix A. For 47 ancient skeletal elements, PCR-MPS was performed using 1.0 to 0.3 ng, 0.1 to 0.3 ng, and less than 0.1 ng of template on 28, 14, and 5 samples, respectively. One nanogram of DNA was always used for each of the 24 petrous bones from Second World War skulls. Figure 2 shows the rDoC of each of the 24 SNPs sequenced in this study, both in the ancient skeletons and in the 24 petrous bones from Second World War skulls (*r*^2^ value among the averages = 0.927). The main sequencing parameters of the bone samples of each skeleton are shown in Appendix A. Differences were found only for the percentage of on-target reads (*p*-value = 0.002), with ancient sk_7 and ancient sk_16 showing the lowest ones (13.6% and 25.6%, respectively).

In addition, it was observed that very low percentages of on-target reads always came from libraries that exhibited the anomalous read length patterns shown in Figure 3.

Appendix A summarizes the comparison of the data between the overall values of the 11 ancient skeletons and 24 petrous bones from Second World War skulls. In agreement with higher levels of degradation, the percentage of on-target reads and the uniformity of the sequencing in ancient skeletons was lower than in Second World War skulls (56.6 vs. 83.3% and 89.9 vs. 99.9%, respectively; *p*-values < 3.8 × 10^−6^).

### 3.3. Genotyping

Out of the 48 libraries built from the ancient bones, 8 were discarded because no read above the 20× threshold was identified by the Converge^TM^ software (version 2.0). The results of the remaining 40 libraries are summarized in Figure 4. In detail, 51/960 markers (5.3%) were below the locus-to-locus threshold values applied in this study, and therefore typed as “NN” (in the Second World War skulls, none of the 576 markers were below the threshold).

No difference (*p*-value = 0.073) was found in the two sample sets for allelic imbalance. The homozygous markers of the ancient samples were flagged for the presence of low covered nucleotides (e.g., below the threshold of 10%) with higher frequency (*p*-value = 0.007). A peculiarity of the ancient samples was the drop-in of a third allele within heterozygous genotypes. As shown in Table 1, a third allele, accounting on average for about 6.4% of the coverage of the marker (min: 2.2%; max: 17.9%), was flagged in eight cases by Converge^TM^ software (see example in Figure A1). This finding (i.e., the presence of more than two nucleotides in the SNP position as well as their ratios) was confirmed independently by the analysis of the BAM files using the homemade bioinformatics workflow [45,46].

The results of the typing of each of the 11 ancient skeletons are shown in Appendix A. For all ancient skeletons (with the exception of sk_16), the data of three to five typings were available, thus allowing a comparison of the genotyping data. This task made it possible to build a consensus profile for all but 20 markers, which were therefore treated as no available data for phenotypic prediction (see below). In addition, this approach also allowed the identification of 28 allelic drop-out phenomena (Figure 5 shows the coverage of the 28 surviving alleles of the markers affected by phenomena). All the consensus profiles determined for ancient skeletons were different.

As shown in Appendix A, full profiles were achieved for the 24 Second World War petrous bones, with an average ratio of allelic imbalances/sample equivalent to one (max: four per sample). No profile showed the identity of the other profiles obtained from the Second World War petrous bones.

Out of the six negative controls (two ENCs and four NTCs), reads above the thresholds were never scored.

### 3.4. Consensus Genotype and Eye/Hair Color Prediction

With the exception of ancient sk_16, for which the data of a unique library were suitable for genotyping, the remaining 10 skeletons provided a consensus profile for 220/240 (91.7%) markers. The phenotypic predictions achieved for all these ancient samples is shown in Table 2 (see Appendix A for details).

In total, a full prediction was achieved for three skeletons (sk_1, sk_3, and sk_5), whereas six skeletons showed a phenotypic prediction for both eye and hair color with different AUC losses because of incomplete genotyping data. For two skeletons (sk_6 and sk_17), no eye color prediction was achieved because of no availability (NA) of data for the rs12913832 marker [3,4,9].

The phenotypic prediction of the 24 Second World War skulls is shown in Appendix A. Out of them, 12 samples yielded a full prediction whereas AUC loss was computed in the remaining 12 samples. Based on the conservative approach adopted (NA data for imbalanced loci), no color prediction was achieved for three skulls, whereas eye or hair color prediction was achieved for one. For both sets of samples, the highest AUC loss was found to occur for brown eye color (0.063 and 0.059) and blue eye color (0.062 and 0.058). AUC loss showed no difference between the ancient skeletons and the Second World War skulls (*p*-value > 0.283).

### 3.5. Unmapped Reads and STR Typing

Eight libraries’ BAM files were checked to verify the presence of non-human DNA sequences (see Appendix A). These eight libraries were selected because of having low percentages of mapped reads (from 0.04% to 23.5%, average = 3.7%, median value: 0.4%), anomalous read length patterns (see Figure 3), and no genotyping results. This analysis revealed the presence of 50- to 80-nucleotide-long sequences in all these libraries, representing 8 to 12% of the unmapped reads. Similarity with *Arthrospira platensis* (34/35; 97%; e-value = 6 × 10^−6^) and *Taylorella equigenitalis* (30/30; 100%; e-value = 2.0 × 10^−5^) was found in five and three libraries, respectively (see Appendix A).

The STR typing of all these samples, performed by PCR-CE, showed full profiles or almost full profiles with the loss of the high-molecular-weight markers, which is in agreement with the degradation level of the samples (see Figure A2) [1]. In addition, the STR analysis confirmed four unique STR profiles (for skeletons 2, 3, 7, and 16) and the biological identity of three skeletal elements of sk_7 and sk_16. No amplicon was scored in the negative controls. Finally, none of these eight samples showed a match with the profiles contained in the elimination database.

## 4. Discussion

High-throughput shotgun sequencing and the analysis of genome-wide data have dramatically improved during the past decade, and they have largely replaced extant PCR-based methods in aDNA analysis [25,26,40]. Briefly, the first step, represented by PCR amplification, is currently omitted, and sequence capture approaches are preferred for generating/selecting barcoded libraries. In the next steps, post-sequencing bioinformatic tools allow the assembly of entire genomes. However, the employment of these new technologies is expensive and is restricted to well-equipped and well-trained laboratories. On the other hand, many PCR-MPS kits have been developed and have become commercially available in recent years [7,51], thus offering the possibility of generating sequencing-based data for a wide forensic public at accessible costs.

In this study, the forensic PCR-MPS approach was applied to ancient skeletal remains and to Second World War bone samples. When dealing with ancient and/or aged forensic samples, stringent precautions need to be adopted for preventing exogenous contamination [36,37,38,39,49]. The employment of sterile materials under suitable working conditions, coupled with the use of blank controls, the replication of the experiments, and/or a multi-sample approach are mandatory, as well as the availability of an elimination database. All these recommendations were followed in this study. Clean negative controls together with a unique eye and hair color phenotype and STR profiles, no match with the elimination database, and C→T transition mediated by the deamination of C to U support the authenticity of genotyping data obtained from aged skeletal remains.

DNA degradation was a common feature of all samples. As expected, it was higher in the ancient skeletons, with 20/47 samples showing un-calculable Auto/Deg ratios, a result that highlights extreme levels of DNA degradation. The availability of sufficient amounts of template DNA (0.325 and 1 ng, respectively), however, allowed generation of libraries of good quality in 40/48 cases for the ancient skeletons and in 24/24 cases for the Second World War skulls, with 94.7% and 100% of the markers suitable for SNP typing. With regard to the eight libraries (16.7%) that yielded no typing, the reasons for this failure should be sought in nonspecific cloning of bacterial DNA sequences because PCR-CE technology yielded STR profiles of good quality from the same DNA extracts (see Figure A2). Similar nonspecific cloning was already reported for other PCR-MPS–based forensic kits used for typing bone remains [50,52]. Thus, the present finding highlights the need of further optimization strategies for library preparation from such samples. For example, purification or isolation of the amplicons before library building could be of help for preventing or limiting this issue [40,53].

The availability of genetic typing data from more than one bone type was used to evaluate the repeatability of the results in 10 out of 11 ancient skeletons. The comparison of the PCR-MPS typing data made it possible to calculate the frequency of the allelic drop-out events, which was up to 5.6%. However, it is likely that the actual percentage of these PCR artifacts is higher because ambiguities could not be resolved for 8.3% of the markers. For example, the marker rs12821256 of ancient sk_6 was typed T/T in four bones, whereas the fifth bone yielded a T/C genotype. In this specific case, either the same allelic drop-out event occurred four times or a single allelic drop-in could be hypothesized (see below). Finally, it has to be emphasized that, in cases of allelic drop-out events, the coverage of the sister surviving allele could be very high. As shown in Figure 5, in fact, the coverage of these surviving alleles was less than 1000× in 28% of the cases, whereas it was even higher than 10,000× (up to 23,572×) in five cases. These results confirm that PCR is an error-prone method for studying aged genomes [1,25,26,45,49] and provide definitive evidence that there is no threshold for a locus call able to protect an expert from the misleading occurrence of allelic drop-out events.

As shown in Table 1, eight loci showed a third nucleotide with a coverage ranging from 2.2 to 17.9% of the entire coverage of the marker (see Figure A1). These “allelic variants” are not reported for any of these bi-allelic markers in the dbSNP web-tool [54]. In addition, they are always consistent with a C→T transition mediated by the deamination of C to U, a base decay described in ancient [37,49] and aged forensic samples [55]. All these data together support the conclusion that they were due to nucleotidic misinsertion rather than to human contamination. Similarly, the deamination of C could cause the drop-in of spurious nucleotides into the markers, thus leading to the background noise observed in 3.8% to 5.3% of the homozygous markers and to (part) of the allelic imbalance phenomena that affected 4.2% to 4.6% of the heterozygous markers typed in this study (see Figure 4). Therefore, unbalanced heterozygous patterns should be always evaluated with caution in samples similar to those employed in this work, and even discarded if no replicate is available for comparison.

Two different strategies were followed for the two sets of samples employed in this study. For each of the 11 ancient skeletons, several different bone types underwent PCR-MPS, whereas only the petrous bone was typed for each of the Second World War skull. The typing data were successfully used for achieving a consensus profile for 16 to 24 SNPs in 10 out of 11 ancient skeletons (see Table 2), and consensus profiles were used for phenotypic prediction [4,56]. Overall, eye and hair color prediction were successfully achieved in 9 skeletons, but with different AUC losses. In the remaining 2 skeletons, eye color was not predicted because of no availability (NA) for data of the rs12913832 marker [4,56]. It has to be noted, however, that in both cases the NA was due to the stringent criteria of the consensus method rather than to a lack of genetic data. Similarly, because 4.2% of the markers were treated as NA in the 24 Second World War skulls, full phenotypic prediction was achieved only for 12 samples, whereas in the remaining 12 skulls different AUC losses were computed. It has to be noted, however, that allelic drop-out events could not be fully ruled out for the 24 Second World War skulls. Drop-outs are expected to occur in PCR amplification of degraded samples even when optimal amounts of sample are used for amplification [1,50,51], and those artefacts were punctually scored in ancient samples, for which 1 ng of template DNA was used for PCR. Therefore, although previous studies performed on up to 80-year-old skeletal remains indicated that use of 1 ng of DNA protects against allelic drop-out risks in the HIrisPlex analysis [27], our data suggest that phenotypic predictions for the Second World War skulls have to be considered prudently.

The Ion Torrent sequencing technology employed in this study is known to be prone to insertion/deletion artifacts [57], whereas the Illumina technology is mainly subjected to misinsertions [58]. In addition, although the error rates generated by the Ion Torrent are higher (≥1%) and DNA library preparation protocols can be more time consuming in comparison with the MiSeq workflow, the time of DNA sequencing is significantly reduced. Therefore, it is certain that each platform offers its own advantages and disadvantages in sequencing [59]. However, it was behind the aim of this work to compare the behavior of different sequencing technologies in analyzing ancient samples. The goal of our work was to show that the Ion Torrent technology (including the automated library preparation and chip loading station) available in our laboratory is suitable for typing ancient DNA samples

## 5. Conclusions

In this study, the forensic PCR-MPS approach was successfully applied for phenotyping ancient skeletons dated from the 3rd to the 18th centuries AD. Following a multi-sample analyses method, in fact, eye and hair color were always achieved even with different AUC loss. Although reproducibility of the results from different bone element types represented the major issue, assuming that appropriate precautions are taken in preventing contamination and that suitable amounts of template are available, our data indicate that ancient samples can be analyzed—at least for a preliminary kit-based attempt—even in the forensic laboratory at standard analytical conditions and facilities that allow the extraction of DNA from skeletal remains samples in separated rooms and hoods. With regard to the Second World War skulls, although optimal amounts of template from the petrous bones were always used for phenotyping, our results advise against a single-test approach [27], thus confirming that replicate analyses remain an irreplaceable tool for preventing or limiting the risk of mistyping.

## Figures and Tables

**Figure 1 genes-13-01432-f001:**
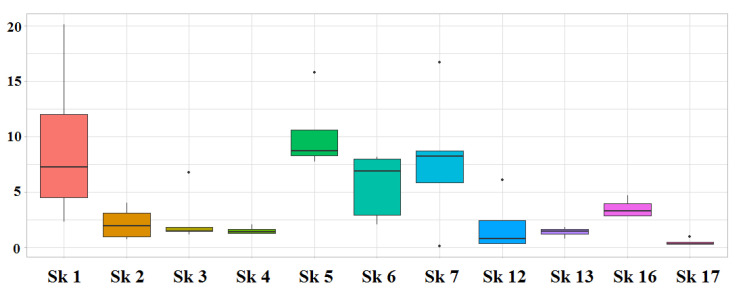
Quantity of DNA extracted from the eleven skeletons. Each “box” spans from the first quartile (Q1, 25% of the data) to the third quartile (Q3, 75% of the data) for each group of data. The bold line inside the box represents the second quartile (50% of the data), i.e., the median for each group of data. The “whiskers” range from the minimum (the smallest value) to the maximum value (the greatest value) of the dataset. The isolated dots outside the whiskers are outliers. *y* axis: nanograms of DNA/gram of bone powder.

**Figure 2 genes-13-01432-f002:**
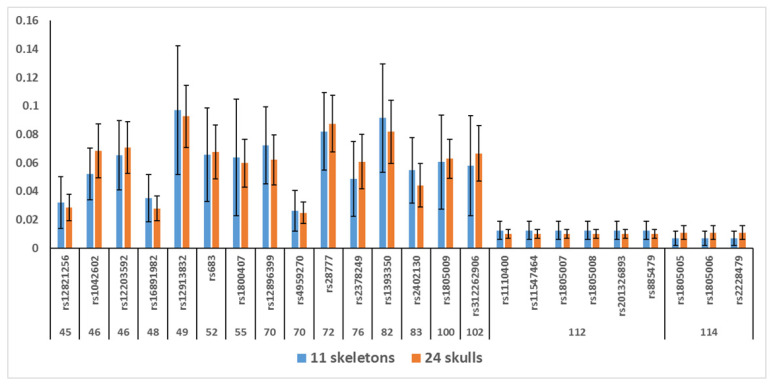
Relative depth of coverage (rDoC, *y* axis) for the two sets of samples (ancient skeletons and Second World War (WWII) skulls). The numbers below the SNP markers (*x* axis) show the length of the sequencing target (in bp). The bars show the standard deviations.

**Figure 3 genes-13-01432-f003:**
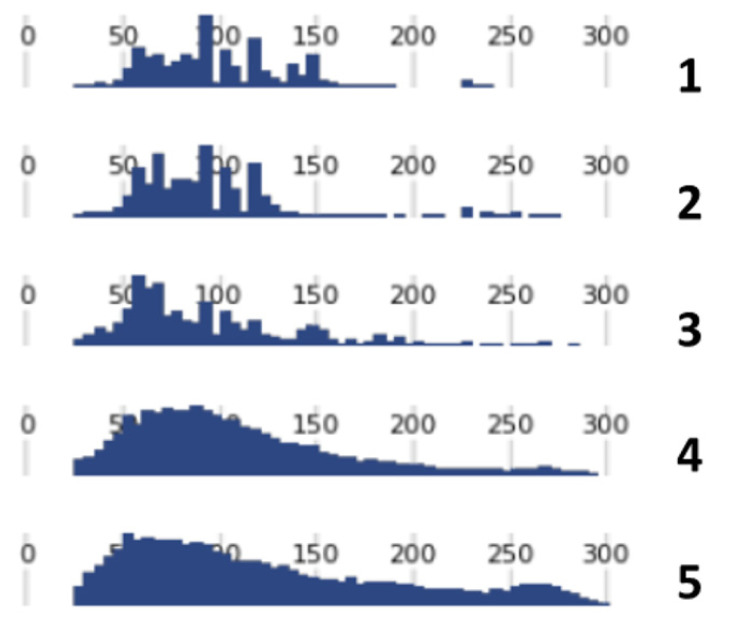
Differences in the length of read profiles in the libraries. 1–3: libraries that yielded about 75 to 85% of on-target reads; 4 and 5: libraries that yielded fewer than 3% of on-target reads.

**Figure 4 genes-13-01432-f004:**
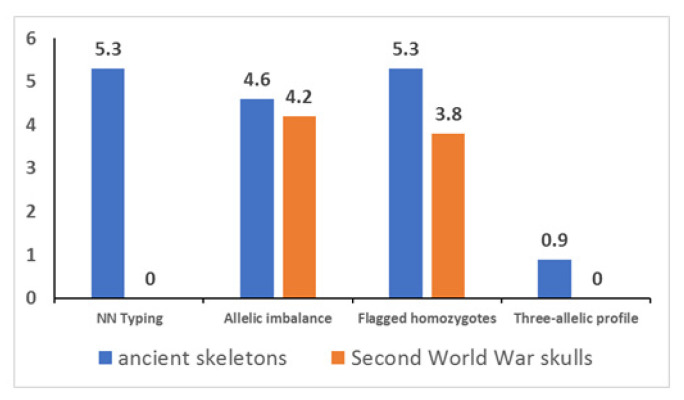
NN typing and alerting flags provided by the Converge software in the two sets of samples (see Appendix A for details on the markers involved). *y* axis: % value.

**Figure 5 genes-13-01432-f005:**
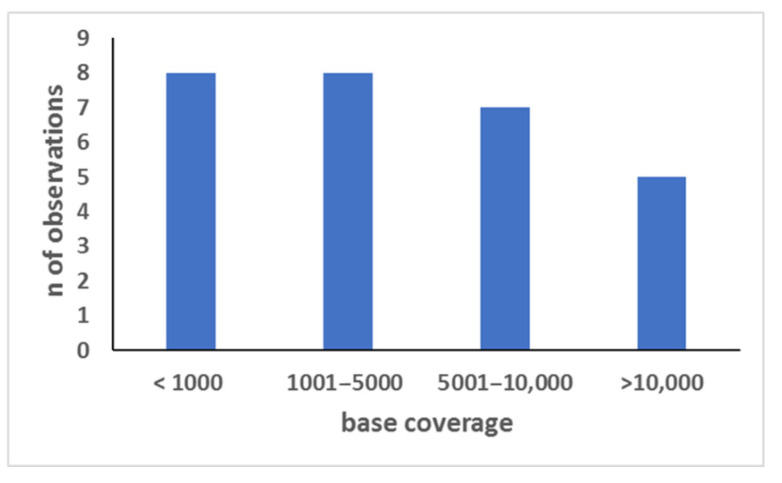
Coverage of the sister surviving alleles in the 28 allelic drop-out events. The observations (n) were pooled according to their coverage into four arbitrarily selected ranges (*x* axis); *y* axis= number of observations.

**Table 1 genes-13-01432-t001:** List of the eight markers showing a tri-allelic profile as found by the Converge software. The dropped-in nucleotide is shown in italics. The base coverage is within the brackets. The last column shows the percentage of coverage of the dropped-in nucleotide.

Skeleton	Sample	Marker	Genotype			%
Sk_1	#141	rs683	A (2576)	C (3930)	*T (707)*	9.8
Sk_1	#142	rs1042602	A (16538)	C (23040)	*T (1079)*	2.7
Sk_4	#97	rs12896399	G (7075)	T (5543)	*A (982)*	7.2
Sk_4	#96	rs12896399	G (12904)	T (6626)	*A (432)*	2.2
Sk_4	#96	rs16891982	C (4217)	G (1943)	*T (199)*	3.1
Sk_6	#59	rs4959270	A (1824)	C (5200)	*T (236)*	3.3
Sk_6	#6D	rs683	A (15494)	C (13807)	*T (1455)*	4.7
Sk_17	#176	rs1042602	C (2943)	A (296)	*T (707)*	17.9

**Table 2 genes-13-01432-t002:** Results from the 11 skeletons. Libraries: total number of libraries. In brackets, the number of libraries discarded as having no marker > 20× of coverage; SNP = number of SNP for which a consensus genotype was achieved (*: for sk_16, the data of a unique library were considered). For both eye color and hair color, the predictions and the AUC loss are provided. und. = undetermined.

Sample	Dating	Libraries	Snp	Eye Color		Hair Color	
				Prediction	AUC Loss	Prediction	AUC Loss
Sk_1	16th	4 (0)	24	blue	0	brown/dark-brown	0
Sk_2	16th	4 (1)	23	brown	0	dark-brown/black	0.148
Sk_3	16th	5 (1)	24	brown	0	dark-brown/black	0
Sk_4	18th	4 (0)	22	brown	0	dark-brown/black	0.013
Sk_5	17th	4 (0)	24	blue	0	dark-blond/brown	0
Sk_6	18th	5 (0)	22	und.	0.727	brown/dark-brown	0.305
Sk_7	18th	6 (3)	16	blue	0.066	blond/dark-blond	0.064
Sk_12	3th	4 (0)	23	brown	0.043	brown/dark-brown	0
Sk_13	3th	4 (0)	23	brown	0	dark-brown/black	0.003
Sk_16	4th	4 (3)	21 *	brown	0	brown/dark-brown	0.02
Sk_17	4th	4 (0)	19	und.	0.812	dark-blond/brown	0.316

## Data Availability

Data is contained within the article or Appendix A.

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
