# Peer review of "Eye and Hair Color Prediction of Ancient and Second World War Skeletal Remains Using a Forensic PCR-MPS Approach"

_genes, 2022, doi:10.3390/genes13081432_

Round 1
Reviewer 1 Report
In this paper, Zupanič Pajnič et al. describe the results of a forensic biology phenotyping NGS-assay to predict hair and eye colours from two series of skeletal remains. Overall, the paper is well written and shows a great proof-of-concept that kits developed for forensic biology can be used for ancient DNA studies.
Introduction
Two suggestions to enrich the reader’s experience:
• I believe that it would be relevant to tell the reader that many forensic biology kits are also available to predict biogeographical ancestry. Together with EVCs panels, theses two techniques can be used to help identify human remains or to provide investigative leads in cold cases like homicide or rape.
• There is no reference to the latest VISAGE consortium panel which are much more powerful than the basic HIrisPlex to predict EVC. I would add this information for the reader’s comprehension of the field at the time of writing.
Results
Figure 1
• I would plot the degradation index on a second Y-axis.
• Did you use the human or the male DNA to report DNA quantification? From my experience, male DNA target are sometimes more reliable for quantification of low DNA quantity. Did you have a big difference between the two?
• Trend in the field is to show a bar graph with individual data points. I would reformat to add individual data points to the graphic. This way, it would be clear that, for example, sk1 probably had a measurement with a very low quantification and another with a higher value resulting in a spread between 2 and 17 for the SD. In such a situation, did you use the mean DNA quant. to calculate the input DNA for library prep? Or did you repeat the quant another time?
Table 1
Table 1 contains a lot of information, but I am bot quite sure what is the take-home message. I would reformat according to the main message: do you want to compare NC between runs, samples, rsSNP. A table, like Table 9 would be OK to show the NN calls and would bring the raw data to the forefront of the paper.
General Comments
Many of the embedded figures are low resolution: it is impossible to zoom in to see the information (especially in the appendices). This should be corrected in the final version.
Author Response
Comments and Suggestions for Authors
Introduction
Two suggestions to enrich the reader’s experience:
- I believe that it would be relevant to tell the reader that many forensic biology kits are also available to predict biogeographical ancestry (BGA). Together with EVCs panels, theses two techniques can be used to help identify human remains or to provide investigative leads in cold cases like homicide or rape.
- There is no reference to the latest VISAGE consortium panel which are much more powerful than the basic HIrisPlex to predict EVC. I would add this information for the reader’s comprehension of the field at the time of writing.
These two issues has been developed in the introduction by adding relevant information on the BGA kits available as well on the VISAGE Consortium panel. In addition, four references has been added. In particular, the following sentenced has been added: “Simillary, to help identify human remains or to provide investigative leads in cold cases like homicide or rape, several PCR-MPS kits allowing bio-geographical ancestry (BGA) determination were developed in the last few years both for Illumina [10,11] and Ion Torrent technologies [12]. The ability of reporting on the EVC, BGA and age based on DNA obtained from biological samples makes the exploration of such markers and the development of new methods meaningful for criminal investigations, and the VISible Attributes through GEnomics (VISAGE) Consortium developed fully optimized and validated the VISAGE appearance and ancestry tool for forensic casework implementation [13,14]''.
Results
Figure 1
We thank the referee for his/her comments to this issue. We agree, in fact, that both the original Figure 1 and the description in the text were not clear at all. Therefore several changes were done (see the revised version).
- I would plot the degradation index on a second Y-axis.
The Degradation Index (Auto/Deg ratio) is reported, for each sample, in Table S2. As already stated in the original discussion, 20/47 ancient samples gave un-calculable ratios because of the lack of amplification of the 249 bp-long Deg target. Therefore, since no further calculation is allowed, data on a second Y-axis can’t be uploaded. Note, please, that a short sentence was added in the revised version to highlight this result.
- Did you use the human or the male DNA to report DNA quantification? From my experience, male DNA target are sometimes more reliable for quantification of low DNA quantity. Did you have a big difference between the two?
As specified in the text, we always used the quantification data as assessed by the 84-bp-long Auto probe. However, we added a short sentence on the Y-specific quantification data (see revised version).
- Trend in the field is to show a bar graph with individual data points. I would reformat to add individual data points to the graphic. This way, it would be clear that, for example, sk1 probably had a measurement with a very low quantification and another with a higher value resulting in a spread between 2 and 17 for the SD. In such a situation, did you use the mean DNA quant. to calculate the input DNA for library prep? Or did you repeat the quant another time?
Figure 1 was changed by using the Stata 16TM software, and median values are shown now. We hope that these modifications make the Figure 1 suitable for the publication of our work.
As already stated in the original manuscript, the quantification was performed, in each DNA sample, by duplicate qPCR tests. This has been added even in the paragraph 2.4 (HIrisPlex sequencing on Ion S5) of the revised version.
Table 1
Table 1 contains a lot of information, but I am not quite sure what is the take-home message. I would reformat according to the main message: do you want to compare NC between runs, samples, rsSNP. A table, like Table 9 would be OK to show the NN calls and would bring the raw data to the forefront of the paper.
Thank you for this comment. This table has been removed and replaced by a Figure showing the percentages of NN typing, allelic imbalance, flagged homozygosity and three-allelic profiles in the two sets of samples (ancient skeletons vs WWII skulls). Only relevant information are reported in the text. Note, please, that part of the original table has been saved as Table S8. This has been done to show the locus-to-locus outcome (and to be in agreement with the data availability statement).
General Comments
Many of the embedded figures are low resolution: it is impossible to zoom in to see the information (especially in the appendices). This should be corrected in the final version.
High quality Figures (TIFF format) are up-loaded in the website of Genes as Supplementary material.
Reviewer 2 Report
The authors applied the PCR-MPS method to generate data for HIrisPlex-S prediction of eye and hair color for DNA from ancient and World War II bone remains. This manuscript was well written and they provided critical data in supplementary materials. Experiments also include essential controls.
I only have a minor comment:
Please explain why Ion S5 was chosen instead of other sequencing methods. The advantage of semiconductor sequencing methods is the high speed and convenience but it is also known notoriously for high error rates. Please explain, perhaps in the Discussion section, how the pros and cons were balanced.
Author Response
Comments and Suggestions for Authors
Please explain why Ion S5 was chosen instead of other sequencing methods. The advantage of semiconductor sequencing methods is the high speed and convenience but it is also known notoriously for high error rates. Please explain, perhaps in the Discussion section, how the pros and cons were balanced.
We thank the referee for his/her suggestions. The following sentences have been added at the end of the discussion: “The Ion Torrent sequencing technology employed in this study is known to be prone to insertion/deletion artifacts [57], whereas the Illumina technology is mainly subjected to misinsertions [58]. In addition, although the error rates generated by the Ion Torrent are higher (≥ 1%) and DNA library preparation protocols can be more time consuming in comparison with the MiSeq workflow, the time of DNA sequencing is significantly reduced. Therefore, it is certain that each platform offers its own advantages and disadvantages in sequencing [59]. However, it was behind the aim of this work to compare the behavior of different sequencing technologies in analyzing ancient samples. The goal of our work was to show that the Ion Torrent technology (including the automated library preparation and chip loading station) available in our laboratory is suitable for typing ancient DNA samples”.
Note, please, that three references have been added as well.